# Selective Sensing of Darolutamide and Thalidomide in Pharmaceutical Preparations and in Spiked Biofluids

**DOI:** 10.3390/bios12111005

**Published:** 2022-11-11

**Authors:** Wael Talaat, Abdelbasset A. Farahat, Reda Mohammed Keshk

**Affiliations:** 1Department of Pharmaceutical Analytical Chemistry, Faculty of Pharmacy, Damanhour University, Damanhour 22514, Egypt; 2Master of Pharmaceutical Sciences Program, California Northstate University, Elk Grove, CA 95757, USA; 3Department of Pharmaceutical Organic Chemistry, Faculty of Pharmacy, Mansoura University, Mansoura 35516, Egypt; 4Department of Chemistry, Faculty of Science, Damanhour University, Damanhour 22511, Egypt

**Keywords:** non-steroidal anti-androgens, organic fluorophore, quenching, dosage form, biofluids

## Abstract

Selective spectrofluorometric sensing is introduced for the analysis of non-steroidal anti-androgens, darolutamide, and thalidomide in pharmaceutical preparations and biofluids. An organic fluorophore, 2,4,8,10-tetramethylpyrido[2′,3′:3,4]pyrazolo[1,5-a]pyrimidine 2 was synthesized in our laboratories by new simple methods to act as a fluorescent reagent for the analysis of the studied drugs. Elemental and spectral analyses were performed to approve the fluorophore structure. The fluorophore possesses a fluorescence at λ_em_ 422 nm when excited at 328 nm. The interaction between the studied drugs and the fluorophore was found to be quenching. The quenching mechanisms were studied and interpreted through the Stern–Volmer relationship. Moreover, the Stern–Volmer constants were calculated for the quenching interactions of both drugs. The introduced method was validated for the estimation of darolutamide and thalidomide in dosage forms, plasma, and urine, offering good percentage recoveries.

## 1. Introduction

Non-steroidal antiandrogens (NSAA) are androgen receptor antagonist drugs that selectively block the effect of androgen hormones such as testosterone and dihydrotestosterone. They are nonsteroids and structurally unrelated to testosterone. The antiandrogenic action of these drugs gives them the merit of treatment of different diseases that are androgen-dependent, such as acne, androgenic alopecia, prostatic hyperplasia, and prostate cancer. Currently, there are two generations of NSAA in the market, the first and second generation [1]. Darolutamide, a second generation NSAA, is commonly used in the treatment of prostate cancer. It is prescribed besides castration in non-metastatic castration-resistant prostate cancer patients. Darolutamide was first synthesized and patented in 2011 [2] and medically approved by the Food and Drug Administration (FDA) and European Union in 2019 and 2020, respectively [3,4]. Darolutamide is *N*-((*S*)-1-(3-(3-chloro-4-cyanophenyl)-1*H*-pyrazol-1-yl)propan-2-yl)-5-(1-hydroxyethyl)-1*H*-pyrazole-3-carboxamide (Figure 1) [5]. There are few reports on the determination of darolutamide; all of them are liquid-chromatographic tandem-mass spectrometric methods [6,7,8,9]. The reported chromatographic technique [6,7,8,9] is very expensive, time-consuming, and requires a large volume of solvents.

(R)-Thalidomide, a non-steroidal antiandrogen drug, is used for the treatment of different types of cancer, such as multiple myeloma and prostate cancer [1]. Thalidomide is 2-[(3R)-2,6-dioxopiperidin-3-yl]-2,3-dihydro-1*H*-isoindole-1,3-dione (Figure 1) [5]. Thalidomide was determined by different chromatographic methods [10,11,12,13].

Our work aims to introduce an effortless, economical fluorescence sensing for darolutamide and thalidomide in pharmaceutical forms and in biofluids. The present investigation is the first fluororimetic analytical method for the analysis of both drugs, darolutamide and thalidomide. There are several fluorescence-based methods for the detection and quantitation of drugs and chemicals [14,15,16,17]. The nucleoside analogues, trifluridine and tipiracil, have been recently determined by fluorimetric sensing [14]. The tricyclic antidepressant, tianeptine, has been analyzed utilizing its quenching effect over the fluorescence of vilazodone [15]. Other fluorimetric methodologies have reported to the analysis of nitroaromatic explosives [16] and some polycyclic aromatic hydrocabons [17]. The proposed method is simple, cost-effective, and not time-consuming.

## 2. Experimental

### 2.1. Materials

Darolutamide was obtained from Angene International Limited, Nanjing, China. Thalidomide was obtained from Sigma-Aldrich, Darmstadt, Germany. Nubeqa^®^ 300 mg tablets, labeled to contain 300 mg darolutamide produced by Bayer pharmaceutical company, Leverkusen, Germany, Thalomide^®^ 200 mg capsules labeled to contain 200 mg thalidomide, produced by celegene pharmaceutical company, Summit, New Jersey, USA and were purchased from a local pharmacy.

The chemicals and solvents were all of the analytical grades. Methanol (99.9%), Chloroform (99.5%), hydrochloric acid, and sodium hydroxide were supplied from El-Nasr chemicals, Cairo, Egypt. Acetonitrile, surfactants like Tween 80 (TW80), cetyltrimethylammonium bromide (CTAB), and sodium dodecyl sulfate (SDS) and potassium dihydrogen phosphate were all supplied by Sigma-Aldrich, Darmstadt, Germany. Orthophosphoric acid (85%) was supplied by S.D. Fine Chemicals Limited. Mumbai, India. Distilled water was supplied by a home laboratory.

Stock solutions of both pyridopyrazolopyrimidine 2, darolutamide, and thalidomide (100 µg/mL) were prepared in methanol. The produced stock solutions were further diluted with the same solvent to obtain the working solutions. The surface-active agents were prepared in an aqueous phosphate buffer adjusted to pH 7.2 to obtain concentrations of 0.012 M for each. The synthesis reagents and solvents were supplied from Sigma-Aldrich Darmstadt, Germany and Fluka, Buchs, Switzerland.

### 2.2. Instruments

MEL_TEMP II apparatus, Perkin–Elmer FT/IR spectrophotometer and JEOL (500 MHz) (Alexandria University, Alexandria, Egypt) were used to record melting points, IR spectra (KBr), and ^1^H NMR spectra, respectively. ^1^H NMR spectra were performed using DMSO-d_6_ and tetramethyl-silane (TMS) as the solvent and internal standard, respectively. Carbon, hydrogen, and nitrogen elements were analyzed in the Cairo University Unit of microanalysis. All the compounds were within ±0.4% of the theoretical values. Mass spectrum was performed on the direct probe controller inlet part to the single quadropole mass analyzer in (thermo scientific GCMS) model (ISQ LT) using Thermo X-Calibur software. Mass spectrum was performed at Al-Azhar University, Cairo (the Regional Center for Mycology and Biotechnology). Silica gel plates 60 F_254_ were utilized for TLC analysis to monitor the reaction pathway and compound purity.

The fluorimetric measurements were recorded using Cary Eclipse fluorimeter (Agilent Technologies, Waldbronn, Germany) with 1 cm quartz cell and monochromator slit width of 1.5 nm. Quinine sulfate (0.01 µg/mL) standard solution was used to check the calibration and linearity of the instrument. The excitation and emission wavelengths of quinine sulfate are 328 nm and 422 nm, respectively.

### 2.3. Synthesis of 2,4,8,10-Tetramethylpyrido[2′,3′:3,4]Pyrazolo[1,5-a]Pyrimidine 2

Method A: Conc HCl (5 mL) was added to a mixture of 4,6-dimethyl-1H-pyrazolo[3,4-b]pyridin-3-amine 1a (1.62 g, 0.01 mole) or N-(4,6-dimethyl-1H-pyrazolo[3,4-b]pyridin-3-yl)formamide 1b (1.9 g, 0.01 mole) and acetylacetone (1.2 g, 0.012 mole) in absolute ethanol (50 mL). The reaction mixture was refluxed for 30 min and left to cool down. The mixture was added to ice-cold water and neutralized with ammonia solution (36%). The formed precipitate was filtered, washed with water, dried, and recrystallized from ethanol.

Method B: To a mixture of 4,6-dimethyl-1H-pyrazolo[3,4-b]pyridin-3-amine 1a (1.62 g, 0.01 mole), acetylacetone (1.2 g, 0.012 mole) in chloroform (50 mL), phosphorous oxychloride (5 mL), and benzaldehyde (1.06 g. 0.01 mole) were added, and the reaction mixture was allowed to reflux for 30 min. Chloroform was evaporated under reduced pressure, and then ice-cold water was added and stirring was continued for 30 min. The formed precipitate was filtered, dried, and recrystallized from methanol.

Pale yellow crystals; yield (92% from 1a, 95% from 1b (method A)), (81% method B); mp: 191–192 °C (lit. mp 190–192 °C [18]); IR (KBr) cm^−1^: 3040 (CH arom.), 2950 (CH aliph.), 1629 (C=C), 1596 (C=N); ^1^HNMR (DMSO-*d*_6_) δ ppm: 2.58 (s, 3H, CH_3_), 2.63 (s, 3H, CH_3_), 2.80 (s, 3H, CH_3_), 2.81 (s, 3H, CH_3_), 6.91 (s, 1H, pyridine proton), 7.32 (s, 1H, pyrimidine proton); M.S (m/z, %): 133 (base peak, 100), 226 (M^+^, 72); Anal. Calcd for C_13_H_14_N_4_ (226): C, 69.03; H, 6.19; N, 24.77; found: C, 68.79; H, 5.99; N, 24.66.

### 2.4. Application Methodology

#### 2.4.1. Setting up of Calibration Graph

Calibration curves for darolutamide and thalidomide determinations were put up by plotting the pyridopyrazolopyrimidine 2 fluorescence intensity versus the darolutamide and thalidomide concentrations. The reagent fluorescence intensity was firstly determined in absence of the drug as follows: 1.25 mL of the fluorophore standard solution (equivalent to 12.5 µg/mL) were moved to a 10 mL volumetric flask and mixed with 2.0 mL of 0.012 M of surfactant, SDS, with pH 7.2. The resultant solution was diluted with methanol till the mark. The fluorescence of the diluted mixture was determined at 422 nm using λ_ex_ of 328 nm.

Aliquots of the darolutamide and thalidomide standard solutions, 1.25 mL of the reagent, 2.0 mL of 0.012 M of surfactant, SDS, with pH 7.2 were added into a set of 10 mL volumetric flasks. The resultant solution was diluted by methanol till the mark to obtain darolutamide and thalidomide concentrations in the ranges from (0.02–0.5 µg/mL) and (0.05–0.5 μg/mL), respectively. The fluorescence of the diluted mixture was recorded at 422 nm after its excitation at 328 nm. The calibration graphs were plotted, and the regression equations were alternatively deduced.

#### 2.4.2. Sensing of Darolutamide and Thalidomide in Pharmaceutical Preparations

The average weight of ten Nubeqa^®^ 300 mg tablets and ten thalidomide-capsule contents were calculated, and the tablets were powdered. An average weight amount was extracted with a sufficient volume of methanol (80 mL) in a volumetric flask of 100 mL. The suspensions were filtered into another 100 mL volumetric flask and the filtrates were diluted to the mark with the same solvent. Aliquots of the previous solution were transferred to 10 mL volumetric flasks and mixed with 1.25 mL of the fluorophore and 2.0 mL of 0.012 M surfactant, SDS, with pH 7.2. The resultant solution was diluted by methanol till the mark to obtain the darolutamide and thalidomide concentration in the ranges from (0.02–0.5 µg/mL) and (0.05–0.5 μg/mL). The emission intensity was determined at 422 nm using λ_ex_ of 328 nm.

#### 2.4.3. Determination of Darolutamide and Thalidomide in Spiked Biofluids

One milliliter of plasma or urine was convoyed into two separate centrifuge tube series and simultaneously spiked with aliquots of both the darolutamide and thalidomide working solutions (0.03–0.5 µg/mL) and (0.05–0.5 μg/mL), respectively. The matrices were acidified with 1.0 mL of 0.01 M HCl and extracted with four milliliters of ethyl acetate (first extract). The ethyl acetate layers were separated (first extract). The raffinates were then neutralized with 1.0 mL of 0.01 M NaOH and re-extracted with ethyl acetate and the organic layers were separated (Second extract). The first and second extracts were evaporated under nitrogen gas till dryness, and the spent solids were dissolved in methanol and treated as under 2.4.1. The first extract was utilized for analysis of thalidomide, and the second one was used for the analysis of darolutamide.

## 3. Results and Discussion

### 3.1. Synthesis of 2,4,8,10-Tetramethylpyrido[2′,3′:3,4]Pyrazolo[1,5-a]Pyrimidine 2

4,6-Dimethyl-1H-pyrazolo[3,4-b]pyridin-3-amine 1a and *N*-(4,6-dimethyl-1H-pyrazolo[3,4-b]pyridin-3-yl)formamide 1b were prepared according to the reported procedure [19]. Reacting compounds 1a or 1b with acetylacetone in the presence of conc. HCl afforded 2,4,8,10-tetramethylpyrido[2′,3′:3,4]pyrazolo[1,5-a]pyrimidine 2 in high yield percent and shorter reaction time compared to the previously reported procedures [18,20]. The formation of compound 2 from 1b instead of 3-acetyl-4,8,10-trimethylpyrido[2′,3′:3,4]pyrazolo[1,5-a]pyrimidine attributed to the hydrolysis of amide linkage of 1b by hydrochloric acid before cyclization to the final product. Compound 2 was also obtained as an unexpected product from the reaction of 1a with benzaldehyde and acetylacetone in the presence of POCl_3_ in high yield percent instead of 3-acetyl-4,8,10-trimethyl-2-phenylpyrido[2′,3′:3,4]pyrazolo[1,5-a]pyrimidine 3 (Figure 1). Although using POCl_3_ as a catalyst afforded the product in shorter reaction time (25 min) compared to conc. HCl (30 min), using of conc. HCl is the most effective as it afforded the product in higher yield percent.

The compound 2 structure was approved by spectral data and elemental analysis. The IR spectra showed the presence of absorption bands at 1596, 1629, 2950, and 3040 cm^−1^ for C=N, C=C, C-H aliphatic, and C-H aromatic, respectively (Appendix A). The structure was also elucidated by examining the compound ^1^H NMR spectrum, which indicated the presence of four singlet signals at δ 2.58, 2.63, 2.8, and 2.81 ppm assigned to four methyl groups at 2, 4, 8, and 10 positions. A singlet signal was observed at δ 6.9 ppm for pyridine proton, and pyrimidine proton was observed as a singlet signal at δ 7.3 ppm (Appendix A). Compound 2 mass spectrum showed a molecular ion peak at m/z 226 (M^+^, 72%) corresponding to its molecular formula C_13_H_14_N_4_ (Appendix A).

### 3.2. Impact of Experimental Variables on the Fluorescence

The synthesized reagent was scanned to examine its photoluminescence and determine the appropriate excitation and emission wavelengths. After excitation at 328 nm, its emission was picked at 422 nm (Figure 2).

Various factors influencing both the fluorescence and quenching of the fluorophore were studied. The studied parameters were a type of diluting solvent, medium pH, reagent volume, surface-active agent type and concentration, and quantum yield.

By examining the emission and excitation shown in Figure 2, a non-mirror image appearance was noticed. This observation is owing to the difference in vibrational structure between the fluorophore ground and excitation states that results from variable ionization constant or pKa values of the two states [21].

Solvents of different polarities such as water, methanol, acetonitrile, and chloroform were used as diluents. The study results revealed that the polarity was not the only factor affecting the fluorescence of the reagent, but the structure of the solvent had a potential influence also. Methanol gave the highest emission intensity. All other solvents gave marked lower intensity than the methanolic solution (Appendix A). Water intervened in hydrogen bonding and so facilitated the electron transfer quenching through the dexter mechanism [21]. Although methanol possesses hydrogen bond formation, it gave higher emission intensity than an aqueous solution, which was due to decreased hydrogen bonding and lower vibrational energy in methanolic solution [21,22]. Cyano and chloro groups in acetonitrile and chloroform was responsible for the reduced fluorescence as they induced electron transfer and intersystem crossing processes, respectively [21].

The influence of medium pH was investigated by recording the emission intensity of the reagent at different pH values ranging from 2–8 (Appendix A). The highest fluorescence intensity was observed at pH 7.2. At a more acidic or more alkaline pH, the intensity tends to decrease. It may be attributed to the partially protonated form of reagent at pH 7.2 fluoresce, while the completely protonated form in acidic pHs and the completely deprotonated form in alkaline pHs do not fluoresce. The effect of pH was studied after the addition of the drugs to the reagent to explore its action of the quenching process and it was observed that pH 7.2 was optimum.

The reagent concentration effect on the fluorescence intensity was studied by using different volumes of the reagent methanolic solution (10 μg/mL) ranging from 0.5–3.0 mL. A volume of 1.25 mL was found to be optimum as it gave the highest fluorescence intensity (Appendix A). Any further increase in the reagent volume resulted in a response decline. Two suggested factors may explain this observation. The first one is deduced from the fluorophore excitation and emission spectra partial overlap, which promotes the energy transfer process by means of absorption of part of the emitted energy by the reagent itself [22] (Figure 2). The other factor is the unequal intensity distribution of the incident light throughout the solution of higher concentration [16]. The effect of the reagent concentration was studied after the addition of the drugs to the reagent to explore its action in the quenching process, and it was observed that 1.25 mL of 10 μg/mL was optimum.

Different types of surfactants, non-ionic, cationic, and anionic surfactants, were tried with different concentrations over the critical micellar concentration (CMC). The CMCs of the surfactants were obtained from the literature, and they were 1.2 × 10^−5^ [23], 1.0 × 10^−3^ [24] and 8.2 × 10^−3^ M [25] for TW80, CTAB and SDS. Only the anionic and cationic surfactants increased the fluorescence intensity, while Tween 80 produces no fluorescence, which may be attributed to the formation of a large number of hydrogen bonds with the reagent that facilitates the electron transfer resulting in quenching through the dexter mechanism.

The increased emission intensity of the SDS and CTAB solutions has two possible explanations. The formed micelles enhance the reagent solubility and protect it from quenching by oxygen through diminishing its entrance [17]. The other one is that the increased viscosity of the medium resists the molecule rotation, increasing the chance of parallelism between the molecule momentum vector and the light electric field [26]. The anionic surfactant SDS possesses a higher quenching effect; hence, SDS was selected for further investigation. This is owed to the SDS negative charge that stabilizes the positive hole on the reagent after the electron donation.

Different concentrations of SDS were studied over the range from 0.0002–0.02 M. It was found that concentrations over 0.01 M gave the highest reagent fluorescence and exert high quenching. A concentration of 0.012 M was selected to improve the method’s robustness (Appendix A).

The quantum yield of the synthesized fluorophore was also calculated utilizing the comparative method [27]. Two graphs of absorbance against the integrated fluorescence intensity of low concentrations of both solutions of the quinine sulfate and the fluorophore were plotted (Appendix A) and the upcoming Equation (1) was applied:(1)$$\Phi_{x}=\Phi_{st}(\frac{G_{X}}{G_{ST}})(\frac{\eta^{2}{}_{x}}{\eta^{2}{}_{st}})$$
where the subscripts *x* and *st*. refer to the solutions fluorophore and standard quinine sulfate, respectively. Φ is the quantum yield, G is the gradient, *η* is the index of refraction. Knowing that Φ*_st_* = 54% [28].

As both solutions of the quinine sulfate and the fluorophore were prepared in the same solvent, in addition to being very diluted, it is expected that the same refractive indices will be revealed. The slope values obtained from Equations (2) and (3) were substituted in Equation (1) to calculate the fluorophore quantum yield and it was 52.6%.
Y_a_ = 8450 X_a_ − 110.2 r = 0.996(2)
Y_b_ = 4445 X_b_ − 47.5 r = 0.995(3)
where the subscripts a and b refer to quinine sulfate and fluorophore, respectively. Y is the integrated fluorescence intensity. X is the absorbance.

After optimizing the experimental variables to obtain the highest fluorescence intensity, the interaction between the studied drugs, darolutamide and thalidomide with the synthesized fluorophore, was found to be a quenching effect (Appendix A). A Stern–Volmer plot for the interaction of the darolutamide and thalidomide (quenchers) and the fluorophore was put up (Figure 3). Straight lines were released indicating a dynamic (collisional) quenching process intervened between the quenchers and fluorophore. To be certain of a dynamic process, the Stern–Vomer plot was constructed at varying temperatures, 25–40 °C (Figure 3). The slope of the Stern–Vomer plot increased as the temperature increased, which makes sure of the dynamic (collisional) process [21]. Stern–Volmer constants K_D_ were determined for the collisional interaction of darolutamide and thalidomide; based on the slopes of Figure 3, they gained the values of 69.4 and 60.8 M^−1^, respectively. The binding numbers were determined from the slope of the modified Stern–Volmer plot [28]. The graphs were plotted as in (Appendix A). The slope (Appendix A) was found to be 1.64, which indicates two binding sites for the quenching process of darolutamide over the fluorophore. The queening process can be explained through two mechanisms: halogen- (chloro) induced intersystem crossing resulting in the fluorescence quenching of the fluorophore, and photo-induced electron transfer, which is expected to be directed from the fluorophore to darolutamide (quencher) as a result of the electron deficiency on the phenyl ring induced by the cyano substituent. These two mechanisms may contribute to explaining the two binding sites of quenching [21]. The slope (Appendix A) was found to be 1.11, which indicates one binding site between thalidomide and the fluorophore and can be explained by the photo-induced electron transfer from the fluorophore to the amide linkage in the thalidomide [21]. The electron transfer process is involved in the excited state of a fluorophore that changes the excited state electron configuration, resulting in energy dissipation (quenching).

### 3.3. Validation

The validation of the present analytical and bioanalytical methods was performed by applying the ICH guidelines [29].

The obtained response of the fluorescence quenching was found to be linear related to the darolutamide concentration over the range (0.02–0.5 µg/mL) for the raw material and (0.03–0.5 µg/mL) for the biofluid sample analysis, where the lower and upper limits of quantitation (LLOQ and ULOQ) were 0.03 µg/mL and 0.5 µg/mL. The steady state plasma level of darolutamide after administration of 600 mg/day was estimated to be 4.79 µg/mL [4], which proves that the determined LLOQ and ULOQ of the method are satisfactory for bio analysis. The determined LLOQ and ULOQ calibration curves were constructed for the determination of the drug in raw material, plasma, and urine with correlation coefficients of 0.999, 0.997, 0.997, respectively, and corresponding regression Equations (4)–(6) were deduced:Emission intensity = −590 C + 520 r = 0.999(4)
Emission intensity = −535 C + 515 r = 0.997(5)
Emission intensity = −545 C + 515 r = 0.997(6)
where C is the concentration of the drug in µg/mL.

The statistical analysis of the obtained regression equations revealed small values of standard deviations of residuals (s_y/x_), slope (s_b_), and intercept (s_a_), which figures out the low scattering points around the calibration curves that prove their good linearity. The calculated values of s_y/x_, s_b_ and s_a_ were 9.7, 8.2, 1.12 and 11.43, 8.8, 1.55 and 11.32, 8.6,1.58 for Equations (4)–(6), respectively.

The method was found to be linear for the estimation of thalidomide over the concentration range (0.05–0.5 μg/mL) in raw material and biofluid samples. The lower and upper limits of quantitation (LLOQ and ULOQ) were 0.05 µg/mL and 0.5 µg/mL. The reported plasma level of thalidomide (200 mg) was 2.4 µg/mL after two hours, which proves the bio analysis satisfaction of the method [12]. Calibration curves were constructed for the thalidomide determination in raw material, plasma, and urine with correlation coefficients of 0.999, 0.998, 0.997, respectively, and corresponding regression Equations (7)–(9) were deduced:Emission intensity = −575 C + 520 r = 0.999(7)
Emission intensity = −515 C + 515 r = 0.998(8)
Emission intensity = −535 C + 515 r = 0.997(9)

The statistical analysis of the obtained regression equations revealed small values of standard deviations of residuals (s_y/x_), slope (s_b_), and intercept (s_a_), which figures out the low scattering points around the calibration curves that prove their good linearity. The calculated values of s_y/x_, s_b_ and s_a_ were 8.7, 8.5, 2.3 and 11.6, 9.8, 2.52 and 10.5, 8.9,2.51 for Equations (7)–(9), respectively.

The accuracy, repeatability (intra-day), and intermediate (inter-day) precision were examined by determining three different concentrations within a day and over three consecutive days. The percent relative error (%Er) and relative standard deviation (%RSD) were calculated for the evaluation of accuracy and precision, respectively. Table 1A,B summarize the results of the analysis and calculations. The obtained small values of %Er and %RSD proves good accuracy and high precision.

The limit of detection (LOD) and limit of quantitation (LOQ) were easily calculated by applying the Equations (10) and (11):LOD = 3.3 Sa/b(10)
LOQ = 10 Sa/b(11)
where Sa is the standard deviations of intercept and b is the slope of the corresponding regression equations [30]. The calculated values of the LOD for darolutamide determination in raw material, spiked plasma, and urine were 0.0063, 0.0095 and 0.0096, respectively. The LOQs for darolutamide determination in raw material, spiked plasma, and urine were 0.019, 0.0289 and 0.029, respectively. The LOD values for thalidomide were 0.013, 0.016 and 0.015 in raw material, plasma, and urine, respectively. The LOQ values were 0.04, 0.049, and 0.047 in raw material, plasma, and urine. The small values of the LOD and LOQ indicate the high sensitivity of the method.

The robustness of the proposed method was studied by inducing small changes in the medium pH, reagent, and surfactant volumes, and the %RSD was calculated for each change. The deliberate changes were in the pH range from 7.0–7.4, the reagent volume from 1.0–1.5 mL and the surfactant volume from 1.75–2.25 mL. The calculated %RSDs were 0.22, 0.47, 0.45 for the pH, the volume of fluorophore, and the surfactant (SDS) changes, respectively. The low %RSD values imply high method robustness.

The matrix effects were checked before bioanalysis by checking the emission intensity of the plain matrix of plasma and urine after extraction with ethyl acetate. Negligible fluorescence responses were noticed, which indicates no matrix effect over the emission intensities.

The selectivity of the sensing method was elaborated by observing the quenching effect of some steroidal anti-androgens such as abiraterone, spironolactone, and non-steroidal anti-androgens such as seviteronel, enzalutamide, and co-administered drugs such as the anti-emetic ondansterone. Abiraterone, spironolactone, seviteronel, and ondansetron show no quenching effect. Unfortunately, enzalutamide exerts a quenching effect on the reagent fluorescence.

The specificity of the present sensing method was also explored by observing the quenching impact of commonly used excipients such as sucrose, lactose, cellulose, carboxymethyl cellulose, and mannitol on the reagent fluorescence. The studied additives showed no quenching effect over fluorophore. These studies support improved selectivity and specificity of the method.

### 3.4. Applications

#### 3.4.1. Raw Material and Dosage Form Analysis

The drug concentration was inversely proportional to the reagent fluorescence intensity. Darolutamide raw material and Nubeqa^®^ tablet were analyzed utilizing the present method and the results revealed good recovery percentages of 100.31 ± 0.34 and 100.24 ± 0.6, respectively, while the recoveries for thalidomide determination in the raw material and thalomide capsules were 99.92 ± 0.28 and 99.9 ± 0.74, respectively Table 2A,B.

#### 3.4.2. Biosensing of Darolutamide and Thalidomide in Spiked Biofluids

The spiked matrices were extracted by applying a liquid–liquid extraction procedure exploiting ethyl acetate as the plasma immiscible extracting solvent. The matrices were acidified with HCl to convert the parent darolutamide to its protonated ionic form, and hence the thalidomide was easily extracted with the ethyl acetate. After the extraction of thalidomide, the raffinates were neutralized with NaOH to obtain the parent darolutamide, which was extracted with the organic solvent. The extracting process was efficient as observed from the mean recoveries of darolutamide determination, which were 100.20 ± 1.40 and 100.15 ±1.30 for spiked plasma and urine, respectively (Table 3), and the recoveries of thalidomide, which were 100.5 ± 1.5 and 99.50 ± 1.35 for spiked plasma and urine, respectively (Table 3).

## 4. Conclusions

We reported new, efficient methods for the synthesis of 2,4,8,10-tetramethylpyrido[2′,3′:3,4]pyrazolo[1,5-a]pyrimidine 2 and confirmed its chemical structure by elemental analysis and spectral data. The synthesized compound was used for the analysis of non-steroidal anti-androgen, darolutamide, and thalidomide in pharmaceutical preparation and in biofluid with high sensitivity and selectivity. The present method can be exploited for the analysis of darolutamide and thalidomide in quality control and bioavailability laboratories.

## Data Availability

Not applicable.

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
