# Peer review of "Selective Sensing of Darolutamide and Thalidomide in Pharmaceutical Preparations and in Spiked Biofluids"

_biosensors, 2022, doi:10.3390/bios12111005_

Round 1
Reviewer 1 Report
This is a thorough study in the area of the analysis of drugs, on the example of darolutamide and the (R) isomer of thalidomide. For this purpose, a photoluminescence indicator was synthesized and well characterized. The beneficial usage of this indicator was demonstrated and validated. The manuscript is fluently written and after minor revision as indicated below the manuscript should be ready for publication.
p. 2, first paragraph: It should be emphasized that it is important to work with the (R) enantiomer of thalidomide as displayed in Figure 1 since the (s) enantiomer is massively teratogenic. This is of highest importance. This is in fact not evident from the first paragraph on p. 2 where the systematic name of thalidomide is given as 2-(2,6-dioxopi-peridin-3-yl)-2,3-dihydro-1H-isoindole-1,3-dione which provides the impression that thalidomide is a racemic mixture of the two stereoisomers.
p. 2, section 2.1. 2nd paragraph, line 3: The acronym CTAB generally designated cetyltrimethylammonium bromide but not cetrimide as indicated in the text. Cetrimide is a mixture of quanternary ammonium compounds, and CTAB is only one component of this mixture. Thus it is unclear if the acronym is not correct of cetyltrimethylammonium bromide was used instead cetrimide.
p. 3, section 2.3., ist paragraph, line 5: The concentration of the ammonia solution is missing.
Author Response
Response letter
The authors are very thankful to the editor and the reviewer for their time, valuable comments, and notes. It is really very useful. We have gone through the comments and answered them to the maximum possible extent. The changes made in the revised manuscript are yellow highlighted. My replies to the comments of the reviewers are described below in a systematic manner. We also carefully checked the format of the revised manuscript to be sure that it meets the journal requirement.
Looking forward to your positive response.
Reviewer #1:
- 2, first paragraph: It should be emphasized that it is important to work with the (R) enantiomer of thalidomide as displayed in Figure 1 since the (s) enantiomer is massively teratogenic. This is of highest importance. This is in fact not evident from the first paragraph on p. 2 where the systematic name of thalidomide is given as 2-(2,6-dioxopi-peridin-3-yl)-2,3-dihydro-1H-isoindole-1,3-dione which provides the impression that thalidomide is a racemic mixture of the two stereoisomers.
The paragraph was corrected to emphasize the use of (R)- thalidomide enantiomer.
- 2, section 2.1. 2nd paragraph, line 3: The acronym CTAB generally designated cetyltrimethylammonium bromide but not cetrimide as indicated in the text. Cetrimide is a mixture of quanternary ammonium compounds, and CTAB is only one component of this mixture. Thus it is unclear if the acronym is not correct of cetyltrimethylammonium bromide was used instead cetrimide.
The paragraph is corrected to indicate that the acronym CTAB is designated cetyltrimethylammonium bromide and not cetrimide.
- 3, section 2.3., ist paragraph, line 5: The concentration of the ammonia solution is missing.
The concentration of ammonia was 36% (inserted in the text)

Reviewer 2 Report
I think that it is an interesting work but I have some doubts:
- A quantitatium limits are indicated. Are these limits enough to consider this method as good?
- About selectivity we indicate that enzalutamide presents quenching effect. So, It is not possible to use this method if we consider that there may be enzalutamide in the sample. Is this correct?
- I would like more information comparing the selectivity and specifity of this methos with that of others used in this field.
- You should improve the quality of some of the figures.
- You should improve the references. I think that references as wikipedia are not adequate.

Author Response
Response letter
The authors are very thankful to the editor and the reviewer for their time, valuable comments, and notes. It is really very useful. We have gone through the comments and answered them to the maximum possible extent. The changes made in the revised manuscript is yellow highlighted. My replies to the comments of the reviewers are described below in a systematic manner. We also carefully checked the format of the revised manuscript to be sure that it meets the journal requirement.
Looking forward to your positive response.
Reviewer #2:
- Quantitation limits are indicated. Are these limits enough to consider this method as good?
The steady-state plasma level of darlutamide after administration of 600 mg /day was estimated to be 4.79 µg/mL [4] which prove that the determined LLOQ and ULOQ of the method are satisfactory for bioanalysis.
The reported plasma level of thalidomide (200 mg) was 2.4 µg/mL after two hours which proves the bioanalysis satisfaction of the method [12].
(Inserted in the text under section 3.3. Validation )
- Regarding selectivity, we indicate that enzalutamide presents a quenching effect. So, It is not possible to use this method if we consider that there may be enzalutamide in the sample. Is this correct?
Yes, that is correct
The selectivity of the sensing method was elaborated by observing the quenching effect of some steroidal anti-androgens like abiraterone, spironolactone and non-steroidal anti-androgens like seviteronel, enzalutamide and co-administered drugs like the anti-emetic ondansterone. Abiraterone, spironolactone, seviteronel and ondansetron show no quenching effect. Unfortunately, Enzalutamide exerts a quenching effect on reagent fluorescence.
- I would like more information comparing the selectivity and specifity of this methods with that of others used in this field.
The sentence "The previous studies support improved selectivity and specificity of the method" was corrected to "These studies support improved selectivity and specificity of the method."
As the authors mean the studies of selectivity and specificity which were made in our lab.
(Inserted in the text under section 3.3. Validation )
To the best of our knowledge, the present method is the first fluorimetric method for the analysis of the studied drugs. The proposed method is simple, cost-effective, and not time-consuming. While the reported methods for the determination of darlutamide and thalidomide are all chromatographic methods which are so expensive, time-consuming, and require a large volume of solvents. By revising the reported methods, the selectivity and specificity were not clearly studied. But it is expected that the chromatographic methods especially those associated with mass spectrometry offer more selectivity and specificity.
- You should improve the quality of some of the figures.
The quality of the figures was improved.
- You should improve the references. I think that references such as Wikipedia are not adequate.
The references are improved.
